# Vortex Beam in a Turbulent Kerr Medium for Atmospheric Communication

**Andrey D. Bulygin** [1], **Yury E. Geints** [1,*] and **Ilia Y. Geints** [1,2]

[1] V.E. Zuev Institute of Atmospheric Optics, Zuev Square 1, 634055 Tomsk, Russia; b.a.d@iao.ru (A.D.B.); geintc.ii17@physics.msu.ru (I.Y.G.)

[2] Faculty of Physics, Lomonosov Moscow State University, Leninskie Gory, 119991 Moscow, Russia

[*] Correspondence: ygeints@iao.ru

**Abstract:** The dynamics of the topological charge of a vortex optical beam propagating in turbulent air while accounting for the cubic nonlinearity is theoretically considered. In a number of examples, we show that the optical beam, self-focusing, manifests itself ambiguously depending on the optical wave power. At near-critical values of beam power, self-focusing leads to enhanced spatial localization of optical vortices and substantial suppression of vortex walk-off relative to the beam axis caused by air turbulence. However, with increasing optical intensity, the modulation instability imposed by cubic nonlinearity becomes significant and contributes jointly with medium turbulence and leads to faster divergence of vortex beams.

**Keywords:** optical vortex; optical turbulence; self-focusing; ultrashort laser pulse

## 1. Introduction

The study of propagation of optical radiation with a nonzero topological charge (TC) in the real turbulent atmosphere is an urgent task because such optical beams are considered to be a basis for development of potentially stable communication links. The study of such vortex beams has been devoted to by a large number of studies (see, e.g., [1–4]). The description of methods for generating vortex laser beams carrying different TCs are discussed in [5]. Practically, beam TC can be measured, e.g., using a CCD matrix equipped with a proper optical setup [6].

Usually, when studying the propagation of vortex laser beams in a turbulent medium, air optical nonlinearity is not taken into account [1]. At the same time, femtosecond atmospheric optics are currently being developed and the question of vortex propagation beams are also actively being studied in the context of high-power ultrafast optical communications [2–4]. As a rule, researchers are interested in the influence of the topological charge on the propagation dynamics of a laser pulse, particularly the effect of the topological charge on the stability and spatial energetic characteristics of optical radiation or the manifestation of some specific effects which are characteristic of nonlinear femtosecond optics with random parameters [4].

At the same time, the topics on how optical medium nonlinearity affects the topological charge of a propagating laser radiation and the possibility of using TC as a communication channel have been less explored. Meanwhile, the vortex beam in a turbulent medium is known to be unstable [7–11], i.e., it splits upon propagation into several vortices with different TCs, while the character of their spatial distribution becomes chaotic, which becomes a big challenge given realistic restrictions on the aperture of a receiving optics. This problem can be overcome to a greater or lesser degree by superimposing phase plates (including focusing different types).

This problem and approaches to its solution in the linear case are presented in [12]. Note, however, that focusing methods have a distance limitation, since the beam waist increases with increases in the focusing length. In turbulent conditions, small distortions

in the initial phase develop rather quickly and may easily disappear before reaching the receiver. The method of using partial receiver apertures for capturing beam orbital angular momentum (OAM) has the disadvantage of a significant loss of information. The use of Gauss–Laguerre modes of different orders can partially reduce the diffraction divergence of the beam, but this method also has known limitations [13,14].

We propose another possible method for the manipulation of the optical vortex dynamics, which relies on the controlled use of nonlinear effects in the propagation medium [4]. The aim of this brief communication is to answer the question on how optical cubic nonlinearity affects the vortex beam propagation in a randomly inhomogeneous (turbulent) medium.

## 2. Materials and Methods

To find a solution to the problem considered, it is necessary to solve the nonlinear Schrödinger equation (NLSE) for an optical pulse envelope propagating in a nonlinear random medium. In this case, one should take into account not only the cubic (Kerr) nonlinearity but also other physical effects associated with higher optical nonlinearities causing pulse transverse collapse arrest and the realization of a pulse filamentation regime.

We consider the NLSE pulse propagation equation in the following form [2]:

$$2ik_0\partial_z U = \hat{h}_k U + \left[\varepsilon_t + \varepsilon_l (UU^*)^{2l} + ik_0\alpha_l (UU^*)^{2(l-1)}\right]U \tag{1}$$

Here, $k_0$ is the wave number at the carrier wavelength (800 nm), "*" denotes a complex conjugate, and $\varepsilon_l$ and $\alpha_l$ are the coefficients for the $l$-th order nonlinearity leading to the self-focusing arrest of the beam. These coefficients account for the physical effects stimulating wave refraction and nonlinear absorption in the self-induced electron plasma, respectively. In Equation (1), the operator $\hat{h}_k$ is introduced, which is responsible for optical wave propagation in a cubic medium:

$$\hat{h}_k = \Delta_\perp + \varepsilon_k UU^* \tag{2}$$

where $\varepsilon_k$ is the coefficient for cubic medium nonlinearity (Kerr nonlinearity) and $\Delta_\perp$ denotes the Laplace operator.

The turbulent phase screen $\varepsilon_t$ is constructed in a regular way by the spectral method [15]. In this case, in order to simplify the calculations the spectral density function of turbulent inhomogeneities, $\varepsilon_t$ is modeled by a step function: $\theta(k - k_g) \cdot \tau$. Here, $k_g$ is the upper cut-off spatial frequency of the pulse spectrum and $\tau$ is a free parameter associated with the amplitude of the turbulence [16].

Next, it will be convenient to consider the problem using normalized coordinates: $r \to r/r_0$, $z \to z/L_r$, $U \to U/A_0$. Here, $r_0$ is the characteristic beam radius for launching the medium, $L_r = k_0 r_0^2/2$ is the Rayleigh length representing the characteristic length of the beam diffraction, and $A_0$ is the initial electric field amplitude.

Let us first consider, for the purpose of the primary analysis, a purely cubic medium. In cylindrical coordinates, the NLSE in normalized coordinates reads as

$$-i\partial_z U = \left(\partial_{rr} + \frac{1}{r}\partial_r + \frac{1}{r^2}\partial_\varphi + \eta UU^*\right)U \tag{3}$$

where $\eta = \varepsilon_k (r_0 A_0)^2$. The solution to this equation can be represented in the following form: $U(r, \varphi, z) = A(r, z)e^{-im\varphi}$, where $m$ is the integer. By substituting this into Equation (3), one obtains:

$$-i\partial_z A = \left(\partial_{rr} + \frac{1}{r}\partial_r - \frac{im}{r^2} + \eta AA^*\right)A \tag{4}$$

From this expression, one can conclude that for a nonzero topological charge, the only physically meaningful variable is the initial beam profile, which approaches zero amplitude at the coordinate origin with a power dependence greater than $r^2$. By using other types

of initial profiles as, e.g., in [3] where the initial optical field with TC $m = 1$ is taken as $A \propto re^{-r^2}$, this leads to an error in the simulation since other types of initial profiles cause field instability and infinite amplitude on the beam axis.

For the problem considered, of TC transmission over some distance in the atmosphere, the issue of the existence of a (quasi-)soliton solution of Equation (4) is of certain interest. This means that, for such solitons, the balance condition must be satisfied:

$$\left( \partial_{rr} + \frac{1}{r}\partial_r - \frac{im}{r^2} + \eta AA^* \right) A = 0 \tag{5}$$

For $m = 0$, the solution to (5) is known as the Townes soliton [8]. However, a question is raised: Are there similar soliton solutions when $m > 0$? In the literature [9–11], only the approximate solutions for large $m$ values are known, but for the most practically important cases of small $m$, apparently no such solutions are reported even in a numerical form.

Optical beams with small TC values are important because for a large $m$, the beam becomes highly unstable upon propagation and, therefore, for controlled TC transmission, it is necessary when considering an effective synthesized beam on a relatively large aperture composed of sub-beams carrying small TC. But even for small vortex beams, their stability in a turbulent medium is in question. Importantly, for the estimations of vortex beam propagation, it is necessary to consider the behavior of the rms (effective) beam radius

$$R_{eff}^2 = P^{-1} \iint UU^* r^3 dr d\varphi \tag{6}$$

where $P = \iint UU^* r dr d\varphi$ is optical power. Hence, the soliton balance condition (5) in accordance with the virial theorem [12] can be presented in the following form:

$$\partial_{zz}\left( R_{eff}^2 \right) = 2 \iint \left( U^* \hat{h}_k U \right) r dr d\varphi \equiv 2H = 0 \tag{7}$$

This expression defines the lower limit of the pulse power, i.e., the so-called critical self-focusing power, for each given beam profile. However, this condition is the only necessary criterion, even in a regular case of a homogeneous medium. Particularly, in a turbulent atmosphere, the development of a local instability takes place, leading to a whole beam breaking up into vortex-like filaments whose dynamics can be studied only numerically. Alternatively, this expression opens the way for partial compensation of beam diffraction by using the optical nonlinearity that enables higher control over the spatial position of phase singularities in the beam profile that carries a nonzero topological charge.

## 3. Results

Consider the initial beam profile in the following form [3,7,9]:

$$U_0(r, \varphi) = A_0 r^m \exp\left\{ -\left( r^2 + im\varphi \right) \right\} \tag{8}$$

with two preset topological charges, $m = 2$ and 4. Vortex beams of such TCs are sufficiently stable against diffraction in a regular cubic medium and are widely used in theoretical simulations. The initial beam radius chosen was $r_0 = 1$ mm. The topological charge $M$ measured on a finite aperture in the experiment can be calculated as a phase integral over a closed loop $C$ in the beam profile, which covers the region of interest [7,9,12]:

$$M = \frac{1}{2\pi} \oint_C (\mathbf{s} \cdot d\mathbf{l}) \tag{9}$$

Here, $\mathbf{s}$ denotes the Poynting vector normalized to the optical intensity:

$$\mathbf{s} = \mathrm{Im}\{U^* \nabla_\perp U\} / UU^* \tag{10}$$

where $\nabla_\perp$ is the gradient operator in transverse variables $(r, \varphi)$. The topological charge $M$, derived according to (9) during beam propagation in the atmosphere, generally can differ from the initial topological charge $m$, which on average should remain constant [12–14]. Furthermore, in the case of laser propagation in a turbulent medium due to instability, the centers of phase singularities can walk off the aperture boundaries.

Figure 1a,b show vortex beam profiles (a) at the beginning of a turbulent optical path and (b) after the 10 m propagation in turbulent atmosphere.

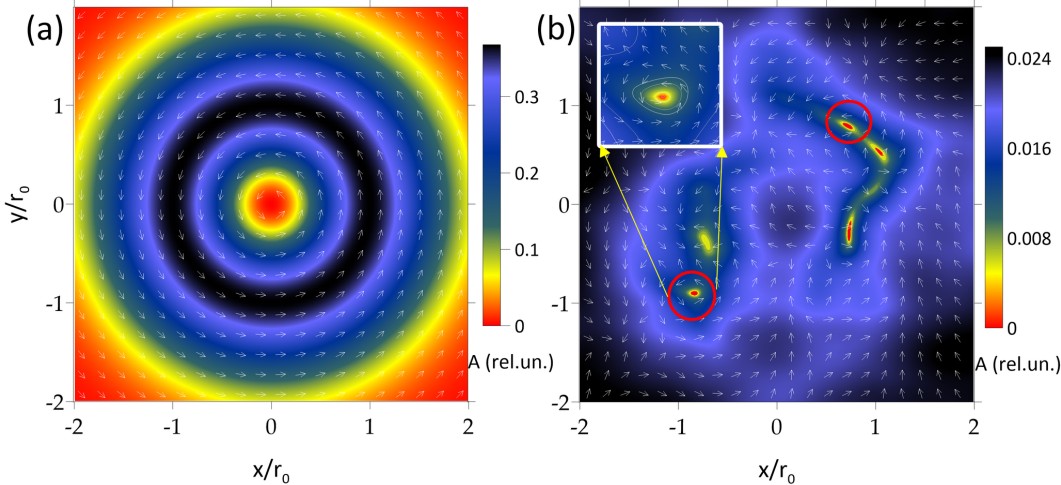

**Figure 1.** Amplitude profile $A$ (color surface) and Poynting vector field **s** (arrows) of a vortex beam with $m = 2$ (**a**) at the optical path beginning and (**b**) after 10 m propagation in turbulent air. Red circles mark the phase singularities. One of them is presented as a close-up picture in the upper left corner in (**b**).

As seen in Figure 1b, the displacement of the singularity centers relative to the beam axis could be substantial. Thus, an undesirable situation is possible when working with a limited optical receiver aperture when one could not gather true information about the beam TC, because only part of the beam will be on the receiver.

In Figure 2, we present the dynamics of the $M$ value calculated within the receiver aperture with the size equal to the initial beam radius (1 mm) for three cases: a linear homogeneous medium without turbulence, turbulent air without cubic nonlinearity, and air with Kerr nonlinearity and medium turbulence.

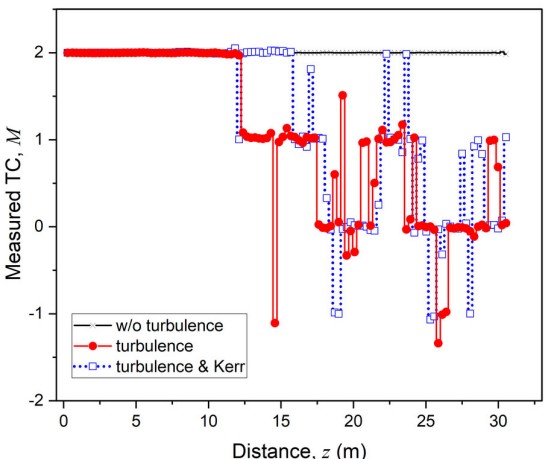

**Figure 2.** Calculated intra-aperture topological charge $M$ of a vortex beam with $m = 2$ propagating in the air in different atmospheric situations.

As seen, the effect of medium turbulence on the TC of propagating a vortex beam manifests itself in the TC changing and even vanishing if measured on the receiver. Moreover, the TC can become fractional or even change its sign (negative). The addition of Kerr nonlinearity partially compensates for the turbulent changes in TC especially at the earlier stage of beam propagation.

## 4. Discussion

For revealing the TC statistical peculiarities, we varied several problem parameters; namely, pulse power, topological charge, and the turbulence spectrum represented by the cut-off wavevector $k_g$. It is worthwhile noting that larger $k_g$ values define smaller-scaled turbulence. The results of our simulations are summarized in Figure 3 for a specific distance $L$, where the initial TC $m$ of the vortex beam becomes lost when measured within a fixed aperture (1 mm). This parameter is defined through the Heaviside function $\theta(x)$ as:

$$L = \left\langle \int_0^{l_0} \theta\left(\frac{M(z)}{m} - 1\right) dz \right\rangle \tag{11}$$

where $\langle \ldots \rangle$ denotes the statistical averaging of data. All the results presented are averaged on twelve independent realizations and normalized to the reference TC range $l_0$ in the case of beam propagation in homogeneous (weak turbulent limit) linear air.

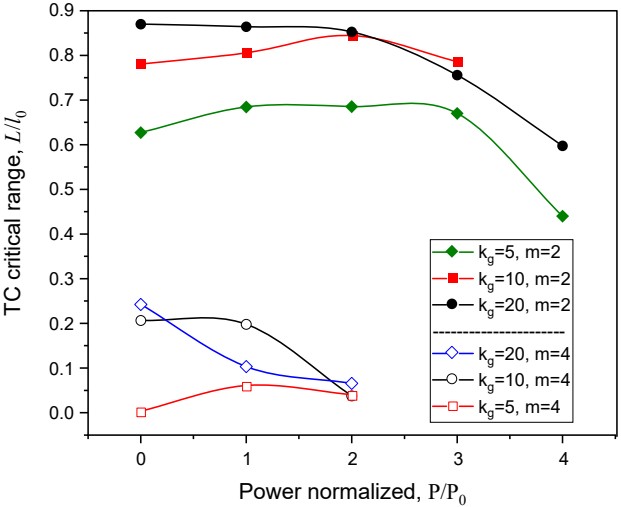

**Figure 3.** Critical range ($L$) for the initial beam TC loss relative to the nonturbulent linear air ($l_0$) at pulse power $P_0 = 3$ GW ($k_g$ is given in cm$^{-1}$).

Obviously, by varying the turbulence scale, laser pulse power, and the value of the beam topological charge from Figure 3, one can reveal a general character of power dependence of the TC critical distance exhibiting a primary increase and subsequent drop in the values of $L$ measured on a limited receiver aperture. Meanwhile, the critical range increases with decreasing initial TC $m$ of the laser beam. At low pulse power, one can see a certain stabilization of singularity center positions near the beam axis assisted by the self-focusing air nonlinearity. Here, for the moderate $m = 2$, the distance $L$ of TC detection in turbulent atmosphere is almost equal to the value in clean air ($l_0$).

However, when the pulse power increases, the influence of air cubic nonlinearity becomes more prominent and the situation reverses which leads to walk-off of the singularities beyond the receiver aperture and $L$ lowering. At the same time, the presence of a peak in the dependences in Figure 3 evidences the manifestation of the stabilizing action of cubic nonlinearity which essentially depends on the turbulence spectrum. The Kerr stabilization

effect of phase singularity positions in a turbulent medium is most pronounced for smaller values of an initial TC of a vortex beam.

When discussing the obtained dependencies, it would be instructive to compare them with the data known from other similar studies. However, generally, it is rather difficult to accomplish, because the obtained results are partial and undoubtedly require generalization and substantiation (both experimental and theoretical). The reason for this is that the problem statement presented in our paper has not ever been explicitly discussed elsewhere thus far.

Nevertheless, it is worth mentioning the study [17], which discusses, first of all, the energy characteristics of the vortex beam and the plasma response of the propagation medium. The dynamics of the effective radius of the beam agrees well with its properties in a cubic medium (see, Ref. [9]) given by Equation (7). However, the relationship between the position of the centers of phase singularities and the effective radius dynamics is still an unexplored question that requires further investigation and discussion.

Another aspect relative to the problem discussed is the pattern recognition, or, in other words, the problem of processing the received optical signal [18]. However, in our study, we do not address the issue of developing a technology for reading information about the topological charge on the receiver. Meanwhile, we do not solve the problem of recognizing a signal distorted by atmospheric turbulence; rather, we seek to understand the conditions when the TC can be delivered to the receiver aperture without losing the necessary information.

## 5. Conclusions

In this paper, we considered the joint effect of medium turbulence and cubic nonlinearity on the possibility of capturing, on a finite receiver aperture, the topological charge of a vortex optical beam propagating in a random inhomogeneous medium (air). From the numerical simulations, we show that the cubic nonlinearity when set at relatively moderate values of initial laser pulse power may be instructive for maintaining the regions of wave phase singularities near the beam axis (within the receiver aperture), counteracting the turbulent divergence. At a relatively high pulse power, the Kerr nonlinearity begins to act along with the turbulence and causes faster beam transverse spreading upon pulse propagation.

Obviously, for more realistic situations, one should consider a broader range of pulse parameters including the initial beam profile, phase distribution, and type of spatial focusing, as well as larger beam scales in real atmospheric conditions. However, the constructive role of cubic nonlinearity makes it one of the promising mechanisms for high-power vortex radiation control for femtosecond atmospheric communications. For this purpose, it is necessary to implement the search for optimal profiles on a wider class of vortex beams and with a more realistic type of medium turbulence meeting with real atmospheric conditions.

**Author Contributions:** Conceptualization, Y.E.G. and A.D.B.; software, methodology, A.D.B.; validation, A.D.B. and Y.E.G.; formal analysis and investigation, A.D.B.; data curation, I.Y.G.; writing—original draft preparation, A.D.B.; writing—review and editing, Y.E.G. and I.Y.G.; supervision, Y.E.G.; funding acquisition, Y.E.G. All authors have read and agreed to the published version of the manuscript.

**Funding:** This research was funded by the Ministry of Science and Higher Education of the Russian Federation (V.E. Zuev Institute of Atmospheric Optics SB RAS).

**Institutional Review Board Statement:** Not applicable.

**Informed Consent Statement:** Not applicable.

**Data Availability Statement:** Data will be made available on request.

**Conflicts of Interest:** The authors declare no conflict of interest.

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
