# Peer review of "Vortex Beam in a Turbulent Kerr Medium for Atmospheric Communication"

_photonics, doi:10.3390/photonics10070856_

Round 1
Reviewer 1 Report
Please see the attachment.

Author Response
Respond to Reviewer#1
N.B. All the revisions are high lightened in green.
- Reviewer: 2. Materials and Methods. Must check Eq. 4, the result of the term…
Authors: Thank you! Corrected.
- Reviewer: 3.Results. The phrases appear again in the second paragraph of page 4. And the figure 2 is not clear on the far right side of the x-axis.…
Authors: Thank you! Corrected.
- Reviewer: 4. Discussion. The simulations for a specific distance L where the initial TC m of vortex beam becomes lost when measured within a fixed aperture (1 mm), it is not clear about the relation which can use some equations.
Authors: Thank you! A short discussion is added.
- Reviewer: References. The format of some references is not uniform.
Authors: Thank you! Corrected.
Reviewer 2 Report
In this paper, the authors investigated the propagation behavior of vortex beam propagating in a turbulent air with accounting for the cubic nonlinearity. It is an important problem in optics. However, I cannot support the manuscript be published in Photonics at presented form.
(1) In Fig. 1, after 10 m propagation in turbulent air, the authors assert that the phase singularity of vortex beam with TC m=2 displaces relative to the beam axis (indicated by the circles in Fig. 1(b)). I don’t agree that the phase singularity moves to the place denoted by the circles. If there is a singularity at the position of circles, the phase in the center of circles cannot be defined. Therefore, zero light intensity is obtained. Apparently, the phase in Fig. 1(b) can be calculated. Therefore, I suggest the authors provide the phase of entire light beam in Fig. 1.
(2) Figure 2 presents the intra-aperture topological charge M of vortex beam with m = 2 calculated by Eq. (9). The physical interpretation of Figure 2 is unclear. For different atmospheric situations, one can always obtain a value of M. The authors should explain the reason of the random change of M denoted by the red and blue lines.
(3) The authors should provide the schematic of vortex beam propagation model in Section 2 so that the entire paper can be more readable.
(4) Please cite the appropriate references in the paper. For example, “This problem can be overcome to a greater or lesser degree by superimposing phase plates (including focusing of different types).” See, line 37 page 1.
(5) Please revise the sentence in line 133 “because only part of the beam because only part of the beam will be on the receiver.”
Author Response
Respond to Reviewer#2
N.B. All the revisions are high lightened in red.
- Reviewer: In Fig. 1, after 10 m propagation in turbulent air, the authors assert that the phase singularity of vortex beam with TC m=2 displaces relative to the beam axis (indicated by the circles in Fig. 1(b)). I don’t agree that the phase singularity moves to the place denoted by the circles. If there is a singularity at the position of circles, the phase in the center of circles cannot be defined. Therefore, zero light intensity is obtained. Apparently, the phase in Fig. 1(b) can be calculated. Therefore, I suggest the authors provide the phase of entire light beam in Fig. 1.
Authors: Thank you for this comment! Indeed, as correctly noted by the reviewer, in the singularity center the optical intensity should be equal to zero. In Fig. 1(b), the minimum field value (actually with zero amplitude) corresponds to the red color of the 2D-diagram. On the color legend, the red color marks a very small range of field amplitudes close to zero value, which is apparently poorly distinguishable in Fig. 1(b). We have revised this figure.
Note, that there are several similar regions in Fig.1(b) of near-zero amplitude, and only two of these regions have the vector field forming a closed vortex. For visual demonstration of the break-up of initial unstable singularity with m=2 into two stable singularities with m=1, we believe this illustration is sufficient. Moreover, if we calculate the exact value of TC along the indicated contours directly following the definition of TC, we obtain the same result.
- Reviewer: Figure 2 presents the intra-aperture topological charge M of vortex beam with m = 2 calculated by Eq. (9). The physical interpretation of Figure 2 is unclear. For different atmospheric situations, one can always obtain a value of M. The authors should explain the reason of the random change of M denoted by the red and blue lines.
Authors: Thank you for this comment! It should be noted, that M is the value that is measured at the receiver aperture if optical vortices fall within this aperture. However, in a turbulent atmosphere, these vortices can go beyond the fixed receiver aperture, or the value of the optical field intensity at the receiver can be lower than its sensitivity, as in the situation without turbulent beam distortions when it is difficult to talk about correct measurement of field characteristics at all. The situation in Fig. 2 is given as a typical example. More valuable results of statistical data processing are given below in Fig.3.
- Reviewer: The authors should provide the schematic of vortex beam propagation model in Section 2 so that the entire paper can be more readable.
Authors: Thank you! We have extended the description of our vortex beam propagation model.
- Reviewer: Please cite the appropriate references in the paper. For example, “This problem can be overcome to a greater or lesser degree by superimposing phase plates (including focusing of different types).” See, line 37 page 1.
Authors: In revised manuscript, as an example illustrating attempts to control a vortex beam by external focusing, we cite the appropriate works.
- Reviewer: Please revise the sentence in line 133 “because only part of the beam because only part of the beam will be on the receiver.”
Authors: Thank you! This is corrected.
Reviewer 3 Report
In this article, the authors discussed the dynamics of the topological charge (TC) of a vortex
optical beam propagating in turbulent air accounting for the cubic nonlinearity that is
theoretically considered. They studied in detail the effect of optical power on the spatial
localization of the phase singularities. Also, claimed that at a particular optical power, selffocusing
leads to enhanced spatial localization of optical vortices and substantial suppression
of vortex walk-off relative to the beam axis caused by air turbulence. Work is interesting and
might find importance in applications like free space optical communication. However, the
introduction section is lacking literature review and their work is purely theoretical and also lacks references which can relate their work to practical situations. A few of my suggestions are:
1. The introduction is too short, my suggestion to the authors is to give a proper
introduction to the optical vortex beam and discuss in detail the techniques used for
the generation and detection of TC in practical situations.
(https://doi.org/10.1515/nanoph-2018-0072,
https://doi.org/10.1016/j.optcom.2020.126710, https://doi.org/10.1063/5.0054885,
https://doi.org/10.1038/s41377-019-0194-2 ). Also, they can put more effort into justifying the need for study.
2. In my opinion, this paper required a more detailed discussion with an experimental
demonstration of well-established techniques using vortex fields for optical
communication through atmospheric turbulence.
(https://doi.org/10.1364/OL.39.004360, https://doi.org/10.1364/OE.26.010494 ).
3. Since the authors have not carried out experimental validation of their study, I suggest
that either they provide some experimental validation to their approach or they can
carry out a detailed comparison with the already published experimental results of
non-linear effects on vortex beams in a turbulent medium with their theory to validate
their approach (http://dx.doi.org/10.1103/PhysRevLett.111.023901,
https://doi.org/10.1016/j.optlastec.2023.109515).
The manuscript can be accepted for publication if the authors wish to modify the manuscript
according to suggestions.
Author Response
Respond to Reviewer#3
N.B. All the revisions are high lightened in blue.
- Reviewer: The introduction is too short, my suggestion to the authors is to give a proper introduction to the optical vortex beam and discuss in detail the techniques used for the generation and detection of TC in practical situations. (https://doi.org/10.1515/nanoph-2018-0072, https://doi.org/10.1016/j.optcom.2020.126710, https://doi.org/10.1063/5.0054885, https://doi.org/10.1038/s41377-019-0194-2 ).
Also, they can put more effort into justifying the need for study.
Authors: The remark is correct. We have included the works suggested by the reviewer in the references and tried to improve the soundness of our study. Additionally, we revised the title of the paper.
- Reviewer: In my opinion, this paper required a more detailed discussion with an experimental demonstration of well-established techniques using vortex fields for optical communication through atmospheric turbulence. (https://doi.org/10.1364/OL.39.004360, https://doi.org/10.1364/OE.26.010494 ).
Authors: We believe that a detailed comparison of the proposed by the reviewer works with our strudy may be useful. However, it is hardly to accomplish within the framework of this short Communication format. Here, we do not address the question of how to develop a technology for reading the topological charge on a receiver aperture, i.e., we do not address the problem of decoding a useful signal. Rather, we pose the question, under what conditions TC can be delivered to the receiver aperture in a randomly inhomogeneous (turbulent atmosphere) environment without loss of information about the TC value. We note that the problem of signal delivery with minimization of information loss only partially relates with the proposed works. However, nevertheless we have extended the reference list and provided necessary comments to these works.
- Reviewer: Since the authors have not carried out experimental validation of their study, I suggest that either they provide some experimental validation to their approach or they can carry out a detailed comparison with the already published experimental results of non-linear effects on vortex beams in a turbulent medium with their theory to validate their approach (http://dx.doi.org/10.1103/PhysRevLett.111.023901, https://doi.org/10.1016/j.optlastec.2023.109515).
Authors: With respect to the works cited by the reviewer, it should be noted that it is hardly to compare them with our results within the framework of this paper. The results of the cited works are well known, have a sufficiently clear physical explanation and have repeated numerical and analytical confirmation, including those in our works. However, these works have very weak relation to the present paper, which is aimed to the study of the possibility capturing field phase singularities on a fixed receiver aperture and evaluating the influence of cubic nonlinearity of the turbulent propagation medium. Therefore, a direct comparison of the results is not possible in principle. At the same time, some indirect relation between the integral beam size and singularity localization seems to be present, but this relation is ambiguous, which, however, is a subject of a further study.
We have included this discussion in the manuscript (Section 4).
- Reviewer: Technical remarks.
Authors: Thank you! The manuscript is additionally proofread.
Round 2
Reviewer 2 Report
The paper can be accepted.